# Hydraulic Modeling and Remote Sensing Monitoring of Floodhazard in Arid Environments—A Case Study of Laayoune City in Saquia El Hamra Watershed Southern Morocco

El-Alaouy Nafia [1,*], Badreddine Sebbar [2,3], El Houssaine Bouras [4], Aicha Moumni [1], Nour-Eddine Laftouhi [1] and Abderrahman Lahrouni [1]

1 Faculty of Sciences Semlalia, Cadi Ayyad University, Marrakesh 40000, Morocco
2 Center for Remote Sensing Applications, Mohammed VI Polytechnic University, Benguerir 43150, Morocco
3 Centre d'Etudes Spatiales de la Biosphère (CESBIO), Université de Toulouse, CNES, CNRS, IRD, UPS, 31400 Toulouse, France
4 Department of Physical Geography and Ecosystem Science, Lund University, Sölvegatan 12, 22362 Lund, Sweden
* Correspondence: nafia.el-alaouy@ced.uca.ma

**Abstract:** Morocco often faces significant intense rainfall periods that can generate flash floods and raging torrents, causing serious damage in a very short period of time. This study aims to monitor wetland areas after a flash-flood event in an arid region, Saquia El hamra Saharan of Morocco, using a technique that combines hydraulic modeling and remote sensing technology, namely satellite images. The hydrological parameters of the watershed were determined by the WMS software. Flood flow was modeled and simulated using HEC HMS and HEC-RAS software. To map the flooded areas, two satellite images (Sentinel-2 optical images) taken before and after the event were used. Three classifications were carried out using two powerful classifiers: support vector machines and decision tree. The first classifier was applied on both dates' images, and the resulting maps were used as input for a constructed decision tree model as a post-classification change detection process.

**Keywords:** flash flood; hydraulic modeling; remote sensing; Saquia El Hamra; Southern Morocco; semi-arid

## 1. Introduction

Floods are the most visible and destructive hydrologic phenomenon in terms of human and economic loss. Typically, flash floods are caused by large amounts of runoff due to short duration and high-intensity rainfall. Floods also lead to environmental and social problems, such as damage to roads, farms, and infrastructures and sometimes pollute surface water resources via the transfer of industrial waste [1,2], creating many health problems. About 20,000 lives are claimed by flash floods annually [3], and, from 1995 till today, approximately 110 million people have been affected by those catastrophes [1,4].

In late October 2016, a flash flood severely damaged the surroundings of the city of Laayoune in the Saquia El Hamra basin in southern Morocco. The country's climate is arid and semi-arid and is prone to destructive floods. Therefore, flood mapping and the determination of the extent of flooded areas is an essential way of creating flood management and prevention strategies (Figure 1).

In the recent years, food hazard assessment has greatly improved, especially with the use of geographic information systems (GIS) integrated with hydrological and hydraulic modeling [5]. The needed hydrological variables can be obtained from a good-quality digital elevation model (DEM), such as catchment shapes, flow directions, slopes, path lengths, and watershed delimitation [6,7]. The monitoring of flood-affected areas resulting from extreme precipitation and changing land use can be helpful for better understanding

flood events [8–10]. Efforts have also been made to integrate some hydrological models with the GIS environment. Most of these models are physically based distributed models, e.g., HEC-HMS, and HEC-RAS. This integration allows the assessment and prediction of the impact of watershed management practices [11–14]. In this work, HEC-RAS and WMS are integrated to the ArcGIS software for the hydraulic modeling of the flash flood events.

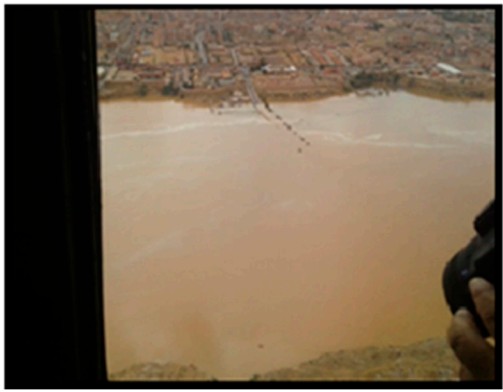 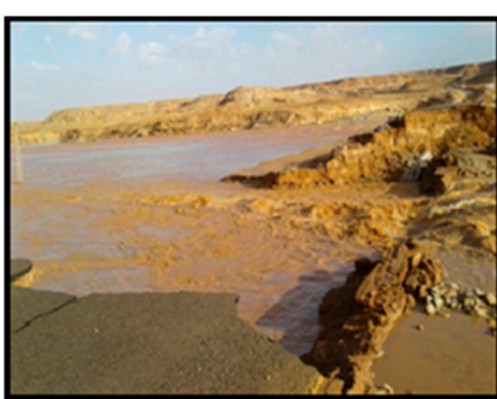

**Figure 1.** The national road N1 linking Laâyoune and Tarfaya submerged on the left, breach opening in the body of the dam embankment by the flash flood.

In the past decades, remote sensing and geographic information systems (GIS) have also opened new opportunities to monitor large areas and create more accurate and valuable flood hazard maps [3,15,16]. High spatial–temporal resolution Sentinel images has been freely provided by the European Space Agency (ESA) for a variety of purposes. In this study, Sentinel-2 optical data was applied for floods analysis. Two techniques were used to extract information from those images: (1) land-cover/land-use classification and (2) a change detection technique. Although no model has proven its superiority, machine learning models have been proven to be better suited for sophisticated flood assessment and have greatly improved flood assessment [1,15,17–20]. In our case, we used support vector machines [21–23] and decision tree models [17–24] for classification and change detection, respectively.

The novelty of this work can be defined in two ways: Using remote sensing technology and hydraulic modeling is an innovative method for examining an event from two perspectives, especially given that remote sensing has made the surveillance of wide areas cost-effective. To summarize, the objective of this study is threefold: (1) to investigate, through modeling, the hydrological regime of the Saquia El Hamra watershed to prevent floods in the future and improve warning systems. The hydrological parameters of the watershed were determined by HEC-RAS and WMS softwares, namely: zone extent, perimeter, slope, basin's average elevation, Gravelius compactness index, Horton shape index, average altitude, drainage density, and concentration time; (2) to approach the problem from the perspective of remote sensing technology to cover the region of interest, to map the affected areas, and to locate the settlements in danger of flooding at the northern part of the city of Laayoune; and (3) to provide a document to support decision-makers in implementing the required protective works and to precisely estimate the financial compensation (e.g., to estimate the destroyed infrastructure and vegetation areas).

## 2. Materials and Methods

### 2.1. Study Area

Laayoune represents the largest city of southern Morocco. This area is located between latitudes 27.00; 27.25 and longitudes −12.92; −13.38 and is bounded by the North Atlantic Ocean to the west. In addition, this city is close to the Saquia El Hamra basin, which is a part of the geological unit of Tarfaya–Laayoune–Boujdour. The watershed of Saquia El Hamra covers an area of 82,000 km. This catchment is characterized by a fairly developed

hydrographic network that drains Oued Saquia El Hamra, which occupies the central part of the Sahara (about 447 km long) (Figure 2).

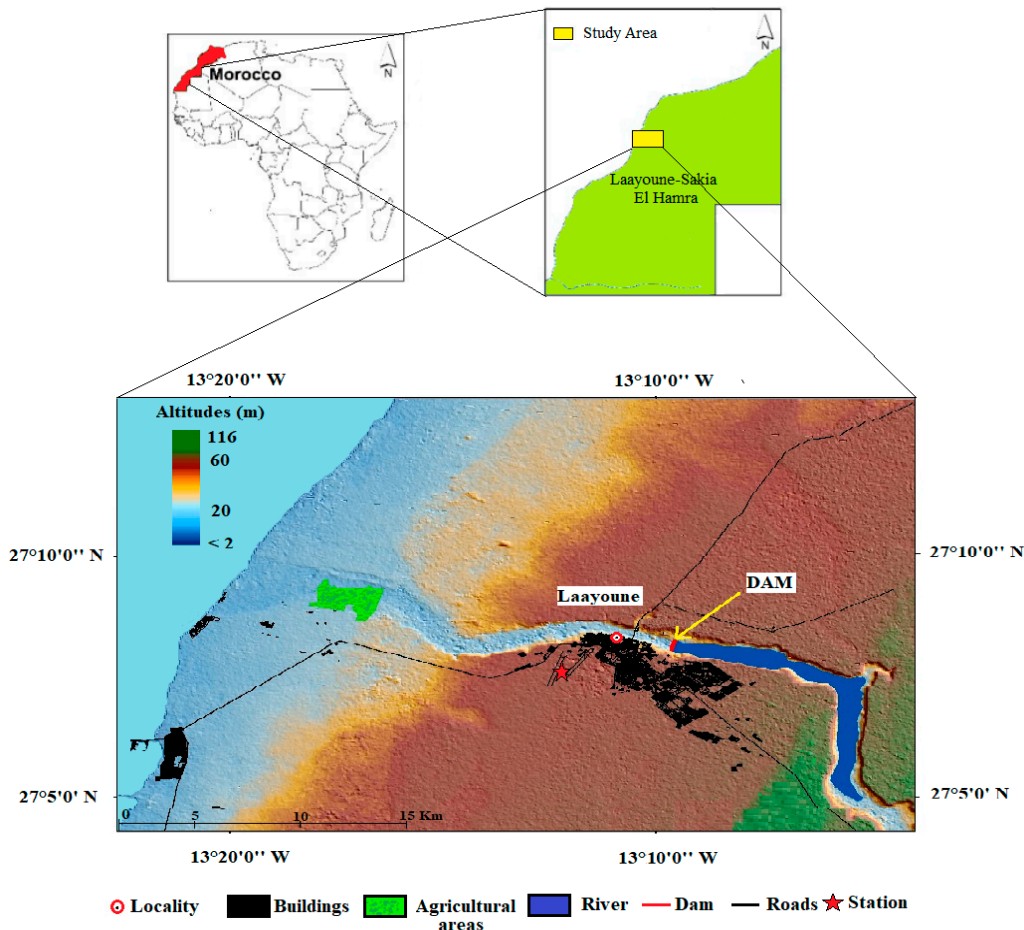

**Figure 2.** Location of the study area.

The climate is generally mild along the oceanic coast, while it becomes more hostile in the interior of the Saharan lands. The rainfall is particularly scarce (Figure 3), and the annual average observed for the past decade is around 60 mm (DNM, 2020).

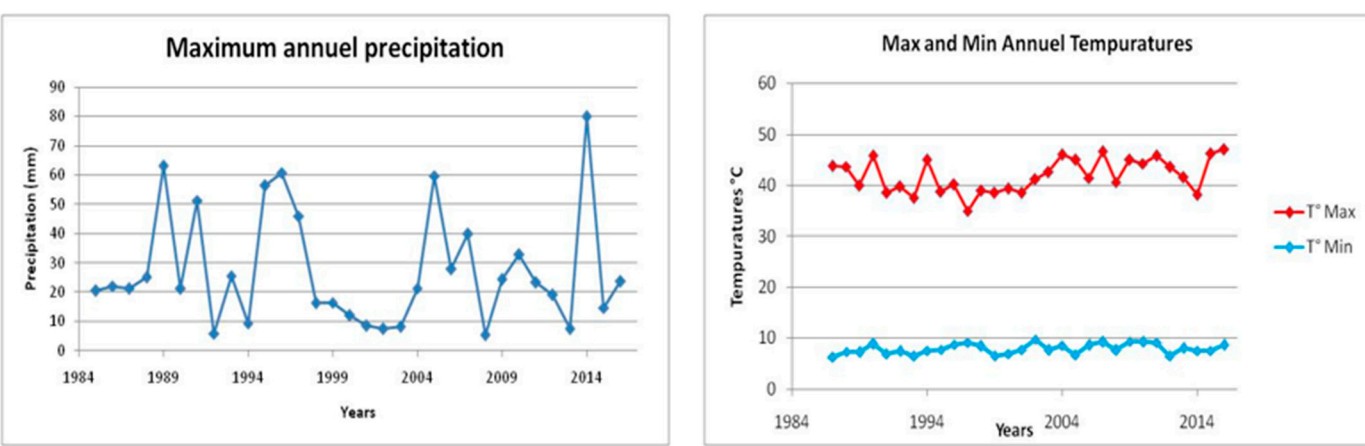

**Figure 3.** Data on precipitation and temperature recorded in the research area from 1984 to 2016.

*2.2. Data Collection*

2.2.1. Hydraulic Model

Due to the insufficiency of precipitation data, the principal hydraulic data employed in the study considered climatic data: maximum annual precipitation and flow data (From 1985 to 2016) provided by the watershed agency of Laayoune Saquia El Hamra Oued Eddahab (ABHSHOD) and the regional directorate of meteorology (DRM) Laayoune Morocco, temperatures (min, max). We used the discharge data recorded on the only station in Laayoune: airport station Hassan I, located at 27.1488° N, 13.2253° W (see Figure 1). Field supplementary measurements from the flooded areas after the event were also received through the ABHSHOD (flow velocity, water height, . . . ). During the years 1993 to 2002, evaporation measurements were taken. The significance and rate of evaporation are primarily determined by the evaporative power of the atmosphere, which, in our case, is measured by a Piche evaporometer. This evaporation represents the atmosphere's evaporative demand.

A DEM (digital elevation model) and the TOPAZ (topographic parameterization) module for simulating the flow direction were integrated in the WMS software for determining the morphological parameters, the boundaries of the watershed, and the hydrographical network.

2.2.2. Remote Sensing Data

The Copernicus Sentinel-2 mission comprises a constellation of two polar-orbiting satellites, Sentinel-2A, launched in 2013, and its twin Sentinel-2B, put in orbit later on, in 2015. They have a wide swath width of 290 km and a high revisit time (10 days at the equator with one satellite and 5 days with the 2 satellites under cloud-free conditions). The MSI optical sensor onboard provides 12 spectral bands (458–2294 μm) from RGB and NIR (near infrared), to SWIR (shortwave infrared), with a high resolution of 10 m, 20 m, and 60 m, respectively [25].

Two Sentinel-2 images were acquired for the present study (Figure 4), one before and one after the flash-flood event (Table 1). These images are freely available, atmospherically, and geometrically corrected from the Theia-Land platform (https://www.theia-land.fr/) accessed on 1 September 2022.

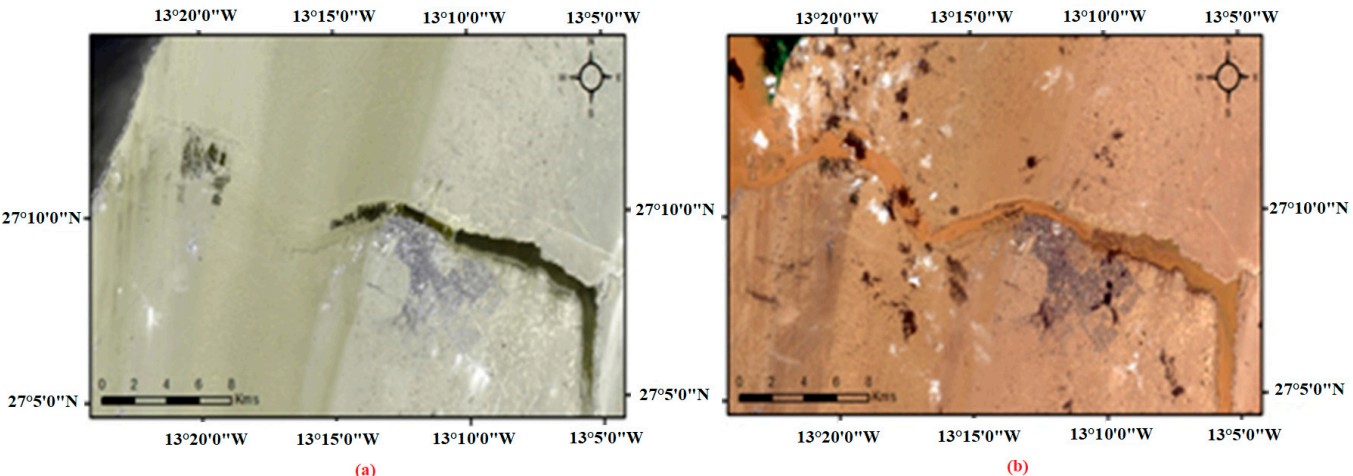

**Figure 4.** (**a**) Sentinel-2 RGB image (bands 2 to 4) eight days before flash flood; (**b**) Sentinel-2 RGB image (bands 2 to 4) one day after flash flood.

**Table 1.** Data images used in this study.

| Satellite | Instrument | Acquisition Date | Use |
|---|---|---|---|
| Sentinel 2 | MSI | 20 October 2016 | One week before flash-flood event; used to calculate reference image |
| Sentinel 2 | MSI | 30 October 2016 | One day after the flash-flood event; used for flood-extent mapping |

### 2.3. Modeling Approaches

The present study is based on a unique coupling between surface hydraulic modeling and high-resolution satellite optical images classification to monitor flooded areas after the flash flood that occurred in Laayoune city in the southern part of Morocco late in 2016. Figure 5 presents the methodological approach followed to achieve this work.

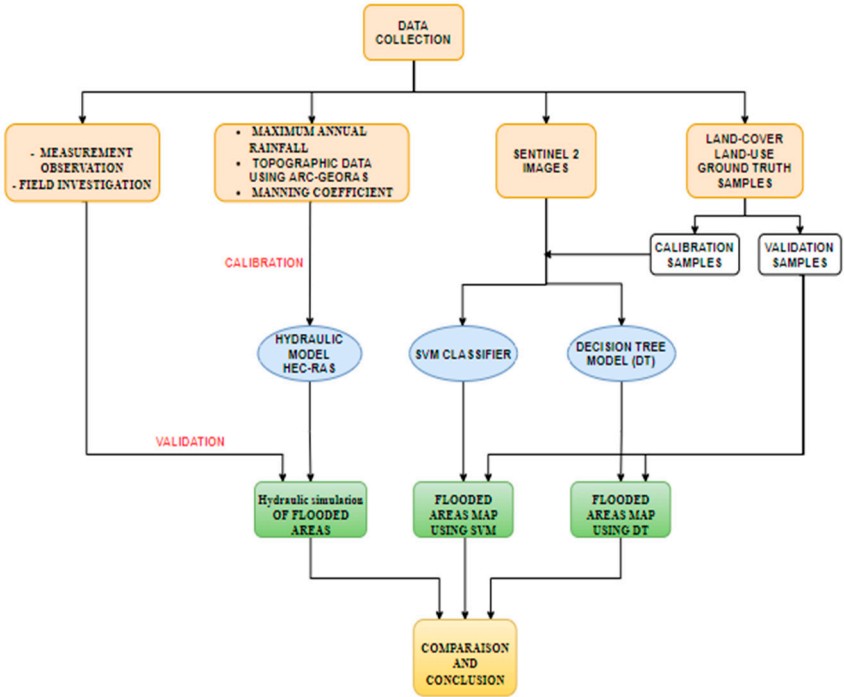

**Figure 5.** Flowchart for the different steps of the methodological approach for flash-flooded area mapping.

### 2.3.1. Hydraulic Model

The delimitation of the watershed is the first step for hydraulic modeling, using the WMS and software developed by the Environmental Modeling Research Lab at Brigham Young University in collaboration with the US Army of Engineers and currently being developed by Aquaveo LLC based in Provo, Utah, USA. It consists of extracting from the DEM the boundaries of the SAKIA EL HAMRA catchment area and its hydrographical network (flow directions). Morphological parameters and physical characteristics of the watershed are also extracted. The methodology for hydraulic modeling is based on coupling WMS and HEC-RAS. The latter is described in the next section.

The daily stream flows were computed using the HEC-HMS 3.4 model, which incorporated the prepared data maps. Meteorological and watershed data were combined to simulate the hydrologic responses.

The hydraulic model implemented in the open-source HEC-RAS software that has been successfully applied and yielded in several international publications and thesis

works [26,27] uses the 1D Saint-Venant shallow-water equations to link water heights and discharges:

$$y_2 + z_2 + \frac{\alpha_2\,v_2^2}{2g} = y_1 + z_1 + \frac{\alpha_1 v_1^2}{2g} + b_e \qquad (1)$$

In this equation, $y_1$ and $y_2$ are water depth in two cross-sections, $z_1$ and $z_2$ are the floor heights of the main channel, $v_1$ and $v_2$ are average velocities of discharge, $\alpha_1$ and $\alpha_2$ are coefficients of mass momentum speed, g is acceleration due to gravity, and be is the head loss of the energy level.

This equation is deduced from the Navier–Stokes equations through simplifications related to the river model. The application of the HEC-RAS model is based on three fundamental steps: (1) Creating, using the ArcGIS tool, the Hec-GeoRAS extension, the Digital elevation model (DEM), and aerial images, the geometrical data of Saquia El Hamra River with the minor and major riverbeds and cross-sections; (2) Applying permanent flow modeling with the Hec-RAS 4.1.0 model, which generates an export file for ArcGIS Hec-Ras, software independent of ArcMap but complementary to the analysis processes; and (3) generating the results of water stain: flood surfaces and depth grids.

Steady flow is a condition in which depth and velocity at a given channel location do not change with time. Therefore, gradually varied flow is characterized by minor changes in water depth and velocity from one cross-section to another [28]. The cross-section sub-division for the water conveyance is calculated within each reach using the following equations:

$$Q = KS_f^{\frac{1}{2}} \;;\text{While } K = \frac{1.486}{n} AR^{\frac{2}{3}} \qquad (2)$$

where K is the conveyance for subdivision, n is the Manning roughness coefficient, A is the flow-area subdivision, R is the hydraulic radius for subdivision (wetted area/wetted perimeter), and Sf is the friction slope.

2.3.2. Satellite Images Classification

Images collected by satellites provide reliable, extensive, and high temporal and spatial resolution data describing surface parameters. Currently, image classification algorithms are the most common mapping method used in satellite imagery. Using this remotely sensed approach, we focus on the classification and identification of the changes in land cover/land use (LCLU) before and after a flash-flood event based on a survey of affected areas, ground truth samples, and temporal images gathered from the Sentinel-2 before and after the event. RGB images were checked visually and showed that reliable discrimination would be based on four main classes: water, vegetation, urban areas, and barren soil. In addition, flooded water was apparent due to the existence of mud. However, to locate vegetation, a spectral index was needed. The normalized difference vegetation index (NDVI) was computed because it is the most used vegetation index [29]. It is calculated using the red (R) and near-infrared (NIR) bands as follows:

$$NDVI = \frac{(NIR - R)}{(NIR + R)} \qquad (3)$$

We used two classification models: the support vector machine classifier and the decision tree model. A brief description of the two methods is presented in the following subsections.

Support Vector Machines (SVM)

SVM are based on statistical learning theory and have the aim of determining the location of decision boundaries that produce the optimal separation of classes [30]. In the case of a two-class pattern-recognition problem in which the classes are linearly separable, the SVM selects from among an infinite number of linear decision boundaries the one that minimizes the generalization error. Thus, the selected decision boundary will be the one that leaves the greatest margin between the two classes, wherein the margin is defined as

the sum of the distances to the hyper plane from the closest points of the two classes [30]. For more details and equations, we invite the readers to refer to the work of [30].

Decision Tree Model (DT)

A decision tree is defined as a connected, acyclic, undirected graph, with a root node, zero or more internal nodes (all nodes except the root and the leaves), and one or more leaf nodes (terminal nodes with no children), which will be termed as an ordered tree if the children of each node are ordered (normally from left to right). A decision tree is built from a training set, which consists of objects, each of which is completely described by a set of attributes and a class label. Attributes are a collection of properties containing all of the information about one object.

To exploit the maximum spectral information, a layer stack of four bands was created for each date: red, green, blue, and NDVI. The resulting raster was fed to the SVM classifier. Using calibration samples from ground truth data, the SVM was trained to detect each class, then compared to decide the classification of each pixel.

The DT model was constructed based on the post-classification maps for both dates before and after the event. This approach is different to the SMV classification technique, as this can be described as a change detection process. The purpose of this procedure is to give more details about the LCLU and to clarify the change that happened in the area. The figure bellow (Figure 6) is an illustration of the change detection DT model constructed (using ENVI 5.3 created by L3Harris Geospatial, based in Boulder, CO, USA).

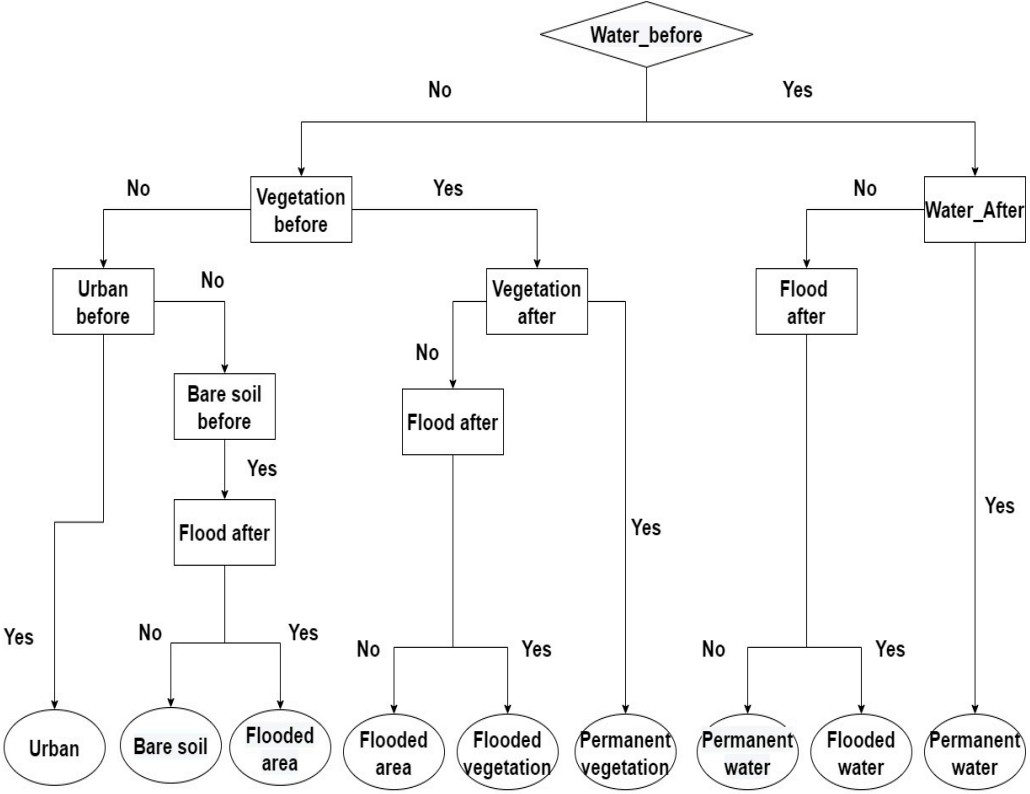

**Figure 6.** Change detection DT model for flooded areas mapping. The mother node is in the rhombus; the decision nodes are in the rectangles; the final classes are in the ovals.

*2.4. Calibration and Validation*

As part of this study, we conducted a field visit to a preliminary investigation survey in order to identify the situation in areas vulnerable to flooding at the level of Laayoune city and the rural commune of Foum El oued after the flash flood, with the help of the team

of the Water Basin Agency of Saquia El Hamra Oued Eddahab. The hydraulic model was validated using cross-sections and water-line profiles realized at the tributary level.

LCLU samples were collected from different parts of the area during field campaigns using a GPS system and labeling the nature of each observed LCLU type in the field using a map of the area. A total of 276 parcel-like polygons of ground-truth data representing all of the classes (resampled into 119 polygons as closer and similar ones were merged) were divided into calibration and validation samples and overlaid on the images (Figure 7). The calibration samples helped training the models, while the validation samples were used in a confusion matrix to assess the evaluation of the performances of the classification, i.e., the overall accuracy and kappa coefficient.

The confusion matrix is calculated by comparing land cover derived from the image against ground-truth land-cover data. Each column of the confusion matrix represents a ground-truth class, and the values in the column correspond to the image's labeling of the ground-truth pixels. The Kappa coefficient, a statistical measure of inter-rater reliability, is calculated as follows [31,32]:

$$Kappa = \frac{N \sum_i^n x_{ii} - \sum(x_{i+}.x_{+i})}{N^2 - \sum(x_{i+}.x_{+i})} \tag{4}$$

where $n$ is the number of rows in the matrix, $x_{ii}$ is the number of observations in row $i$ and column $i$, $x_{i+}$ and $x_{+i}$ are the marginal totals of row $i$ and column $i$, respectively, and $N$ is the total number of observations.

The overall accuracy is calculated by summing the number of correctly classified values and dividing by the total number of values.

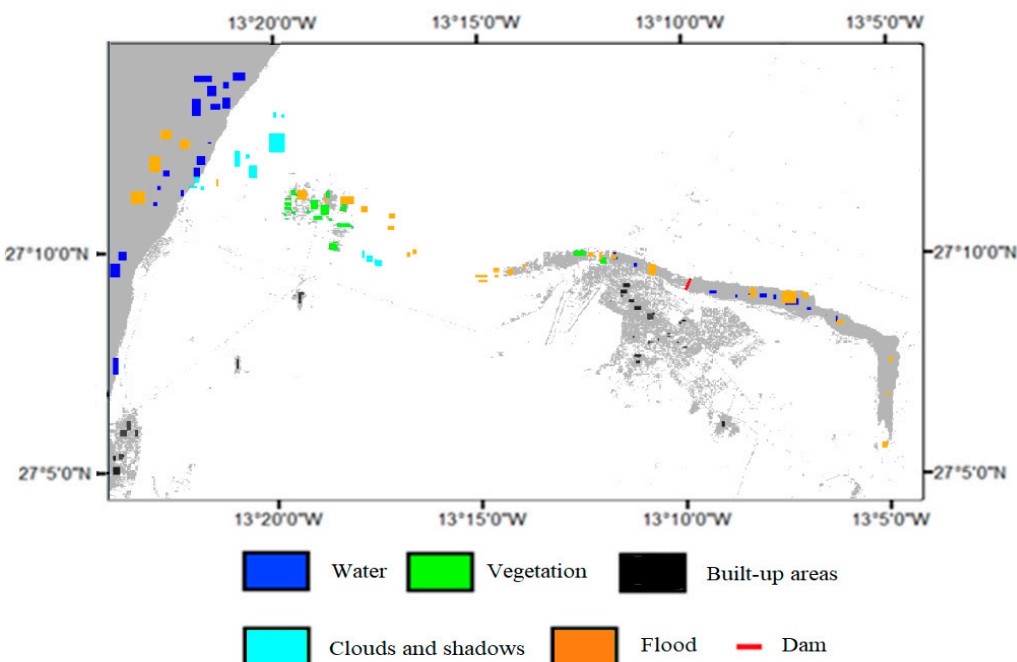

**Figure 7.** Ground-truth data collected during field surveys.

## 3. Results and Discussion

### 3.1. Hydraulic Modeling

A physically based study of the watershed has the purpose of determining the geometrical characteristics of the basin, which covers an area of about 52,000 km$^2$ and has a perimeter of 2033 km. The Gravelius compactness index (Musy 1914), which further decides the relation between the basin's shape and the flow behavior equals 2.4, which means that the basin is elongated. The Horton index indicating the relation of the basin's mean width to the principal water stream's length (221 km) is 0.05, which also indicates that the basin is elongated (KH less than 1). The hypsometric map shows a mean, min, and max altitudes of 293.45 m, 6 m, and 621 m, respectively. The knowledge of slope indices is of great importance, as the water flows more when the slopes are important; it is thus in mountains, we observe, for a given downpour, more important floods than in the plain where the slopes are much lower. The mean, min, and max slopes observed are 0.03 m, 0 m, and 35 m, respectively. The concertation time Tc, defined as the time of the most distant water droplet to reach the outlet, is about 21.3 h.

The highest amount of evaporation occurs in August, the hottest month of the year, and the lowest amount occurs in December. The annual mean evaporation value is approximately 158.99 mm (Figure 8).

An estimate of the periods of return of the extreme values of rain can be obtained by adjusting the laws on pluviometric data. Using the Hyfran-Plus (Hyfran, 1998) and several statistical distributions, such as normal, log-normal, Gamma, Weibull, and exponential, to validate the results, the rainfall depth for various return periods was determined (5, 10, 25, 50, and 100 years). According to the analyses of the graphs of the adjustment of the statistical laws from the data of annual maximum precipitations from a series recorded at the Laayoune rainfall station over a period spanning 31 years (1985–2016) (Figure 9), the exponential distribution is the law best-adapted to the data for analyzing the periods of return, followed by the normal distribution; the other distributions have poor adjustment.

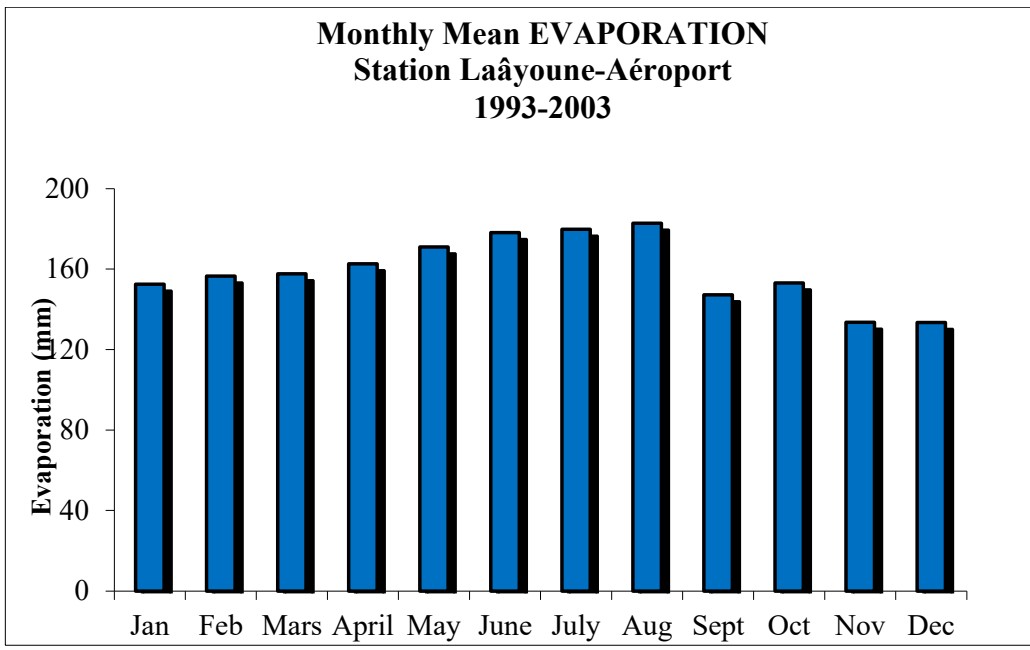

**Figure 8.** Monthly mean evaporation recorded for 1993 to 2002.

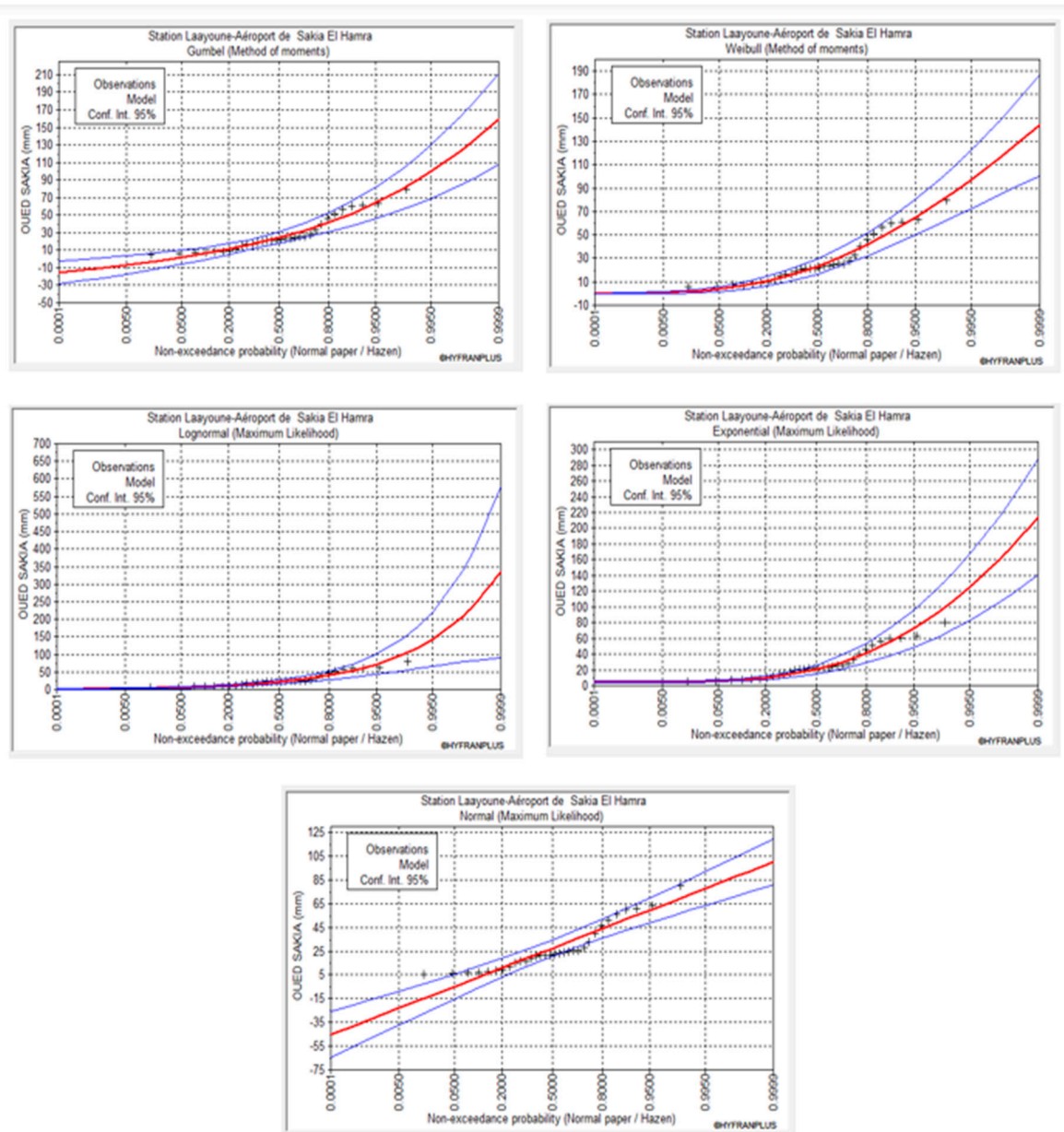

**Figure 9.** Saquia El Hamra's probability distribution curve based on the normal, log-normal, Gamma, Gumbel, and exponential distributions.

The following Table 2 displays the precipitation estimate along with its respective exponential law confidence intervals for various return times.

**Table 2.** The results at different return periods of extreme rainfall events.

| Return Period (Years) | Precipitation (mm) | Confidence at 95% | |
|---|---|---|---|
| 100.0 | 153 | 95.7 | 211 |
| 50.0 | 127 | 83.6 | 171 |
| 20.0 | 96.7 | 67.8 | 126 |
| 10.0 | 75.6 | 55.7 | 95.4 |
| 5.0 | 56.1 | 43.4 | 68.8 |
| 2.0 | 31.7 | 25.6 | 37.9 |

The stream flows were computed using the HEC-HMS 3.4 model, which incorporated the prepared data maps. Meteorological and watershed data were combined to simulate the hydrologic responses. The adjustment of laws to hydrological data yields an estimate of the return periods of extreme flow values, which is crucial for planning, forecasting, and protection activities. The results of the frequency analyses of samples of the maximum annual flows recorded at the Laayoune hydrometric station during the selected 31-year chronicle between 1985 and 2016. The statistical adjustments of each Qmax sample are performed in accordance with various distributions (log-normal, Gamma, Gumbel, and exponential) to identify the distribution that best fits the sample under consideration. Figure 10 depicts the empirical and theoretical probabilities of the distributions of the various statistical adjustments to the maximum daily flow data. We observe that the law best suited to the data for analyzing the return periods is the log normal law, followed by the Gamma law; the other distributions do not fit the data well.

After selecting the law best suited and adjusted to the maximum annual flows of the Laayoune station over a 31-year period, the estimated return periods by this law are shown in the table below (Table 3).

**Table 3.** Estimation of the return period by the normal Log law at the Laayoune station.

| Return Period (Years) | Qmax (m³/s) | Confidence at 95% | |
|---|---|---|---|
| 100.0 | 1570 | 1491.5 | 1644.575 |
| 50.0 | 1070 | 1016.5 | 1120.825 |
| 20.0 | 608 | 577.6 | 636.88 |
| 10.0 | 366 | 347.7 | 383.385 |
| 5.0 | 198 | 188.1 | 207.405 |
| 2.0 | 61.4 | 58.33 | 64.3165 |

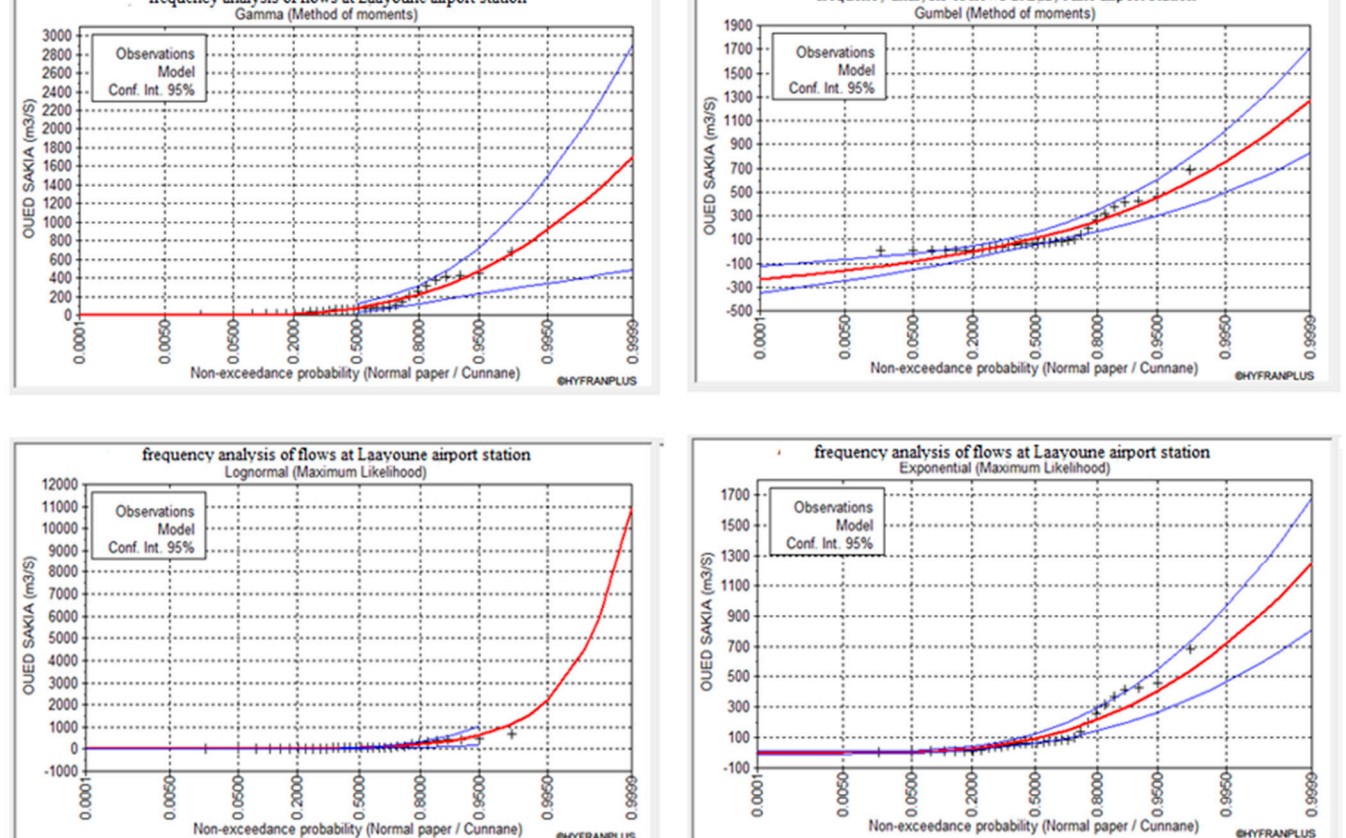

**Figure 10.** Adjustments of the static laws on the annual maximum daily flows.

As a conclusion, the Log normal rule is optimal for forecasting yearly maximum flows, while the exponential law is optimal for forecasting annual maximum rainfall.

There are many outputs available in the HEC-RAS that may be analyzed after simulation such as: cross-sections and flow directions (Figure 11), water surface profiles, general profile plot, rating curves, and property plots. We are mostly interested in the 3D cross-section (Figure 11) in order to be able to compare it with the results from the remotely sensed monitoring of the event.

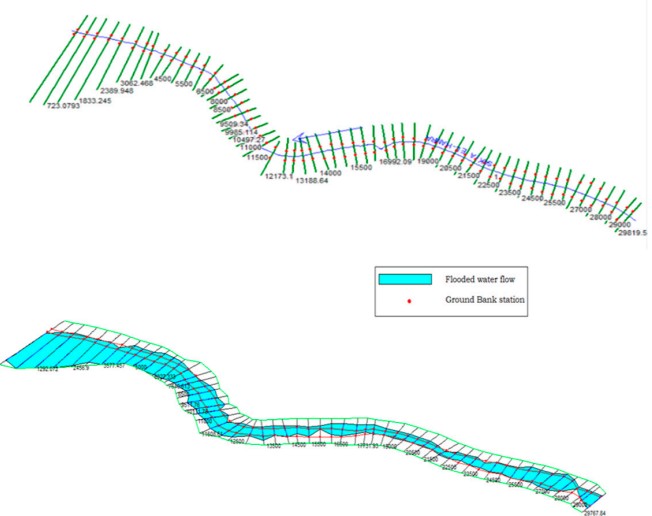

**Figure 11.** Cross-sections and flow direction. The axis of the stream in blue, the sections in green and the banks in red.

Figure 12 shows the flood extent of the studied event, and Figure 13 depicts the flood propagation outcomes for various return periods. Arc GIS uses the results of the hydraulic simulation to produce maps of the floodplain and its limits for various return periods (Q20, Q50, Q100). The results showed that, during flash floods with known flows, the water level can reach up to 13 m, with high flow velocities flooding hundreds of hectares of the surrounding plains at the northern part of the city of Laayoune and agricultural lands near Foum El Oued.

The passage of this exceptional flood caused the wadi's waters to overflow onto the crest of the Saquia El Hamra dam, resulting in the degradation of the "downstream slope" and the opening of two 100 m-long breaches in the dam's body at the level of the wadi's minor bed.

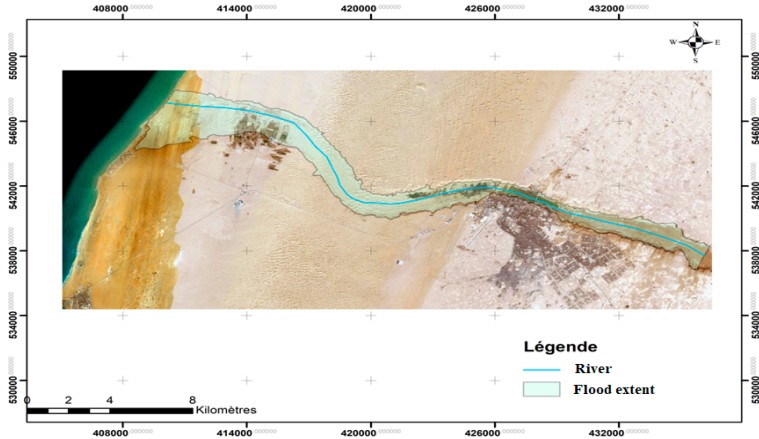

**Figure 12.** Mapping of the extent of the floodplain for the 2016 flash flood.

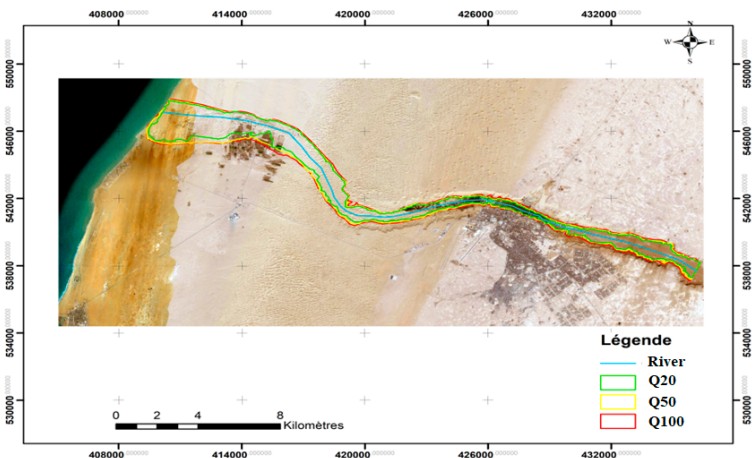

**Figure 13.** A representation of the floodplain for (Q100, Q50, Q20).

Based on the flood propagation map, a hazard map depicting flood-prone areas at the level of the simulated section was generated in ArcGIS (Figure 14). There are tens of hectares of threatened land throughout the entire floodplain. The conclusion of this simulation study reveals the following flood-related issues:

■  At the city of Laayoune's level:

- The destruction of current infrastructure (the national road, N1, linking Laayoune and Tarfaya and the dyke of the Sakia El Hamra dam);
- The destruction of the banks;
- A few homes sustained damage, as well as residents of neighboring communities on the left bank of the Oued (douar Lamkhaznia).

■  At Foum El Oued:

- The inundation of agricultural lands;
- The inundation of the cornice;
- A few houses sustained damage as well.

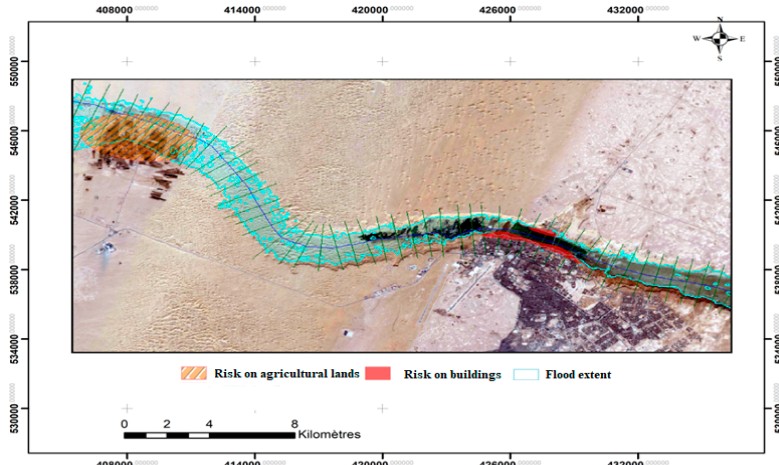

**Figure 14.** Hazard map of the study area.

### 3.2. Remote Sensing Mapping

The resulting map for the first-date classification using the SVM classifier (Figure 15) shows four dominant classes that give a detail about land cover/land use in the area before the flashflood. Three main classes are distinguishable here: (1) seawater around the coastal areas (in the west) and stored dam water (in the east); (2) vegetated areas in farms next to

the sea, in the central zone, and alongside the river; and (3) built-up areas, mainly Laayoune city and port on the coast.

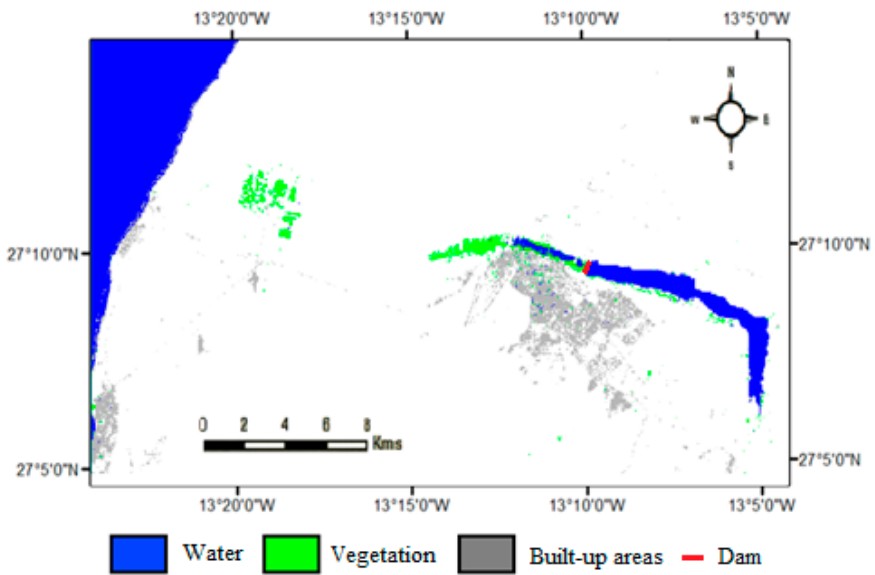

**Figure 15.** SVM classifier's results for the date before the event (20 October 2016).

This first map shows a great match when confronted with ground-truth samples, which is mainly due to the choice of three sufficient and distinguishable classes. The situation in the dam appears to be normal and usual amounts of water stored with small evacuations. The overall accuracy reached 94.41% and a Kappa coefficient of 0.91.

One day after the flash flood, the resulting map using the same classifier (SVM) is shown in the figure below (Figure 16). Two other classes were added for this date, the flooded areas then the clouds and shadow, as, at the acquisition time of the satellite image, some clouds were passing by.

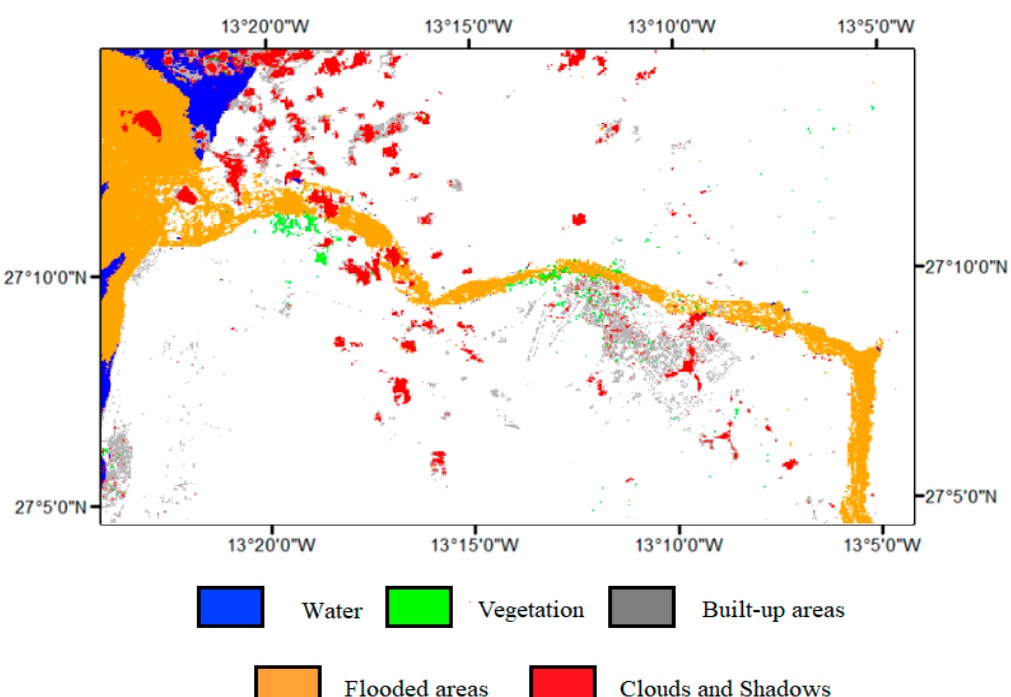

**Figure 16.** SVM classifier's results for the date after the event (30 October 2016).

The overall accuracy and kappa coefficient are 87.31% and 0.82, respectively. The presence of clay in flooded water helped to distinguish the flooded areas, and the reflectance in the visible range and specifically the blue, red, and green spectra were enough to detect it. The added NDVI band separated the vegetation from the other classes and made the decision rule easy for the classifier. This map shows the impact that the flash flood had on the area; the overflow surpassed the dam and affected some neighboring buildings and half of the farmed zone before discharging into the Atlantic Ocean at the outlet called Foum El Oued.

For change detection, we used the decision tree classifier as a post-classification change detection tool (Figure 17). It has rarely been used in the literature for this purpose.

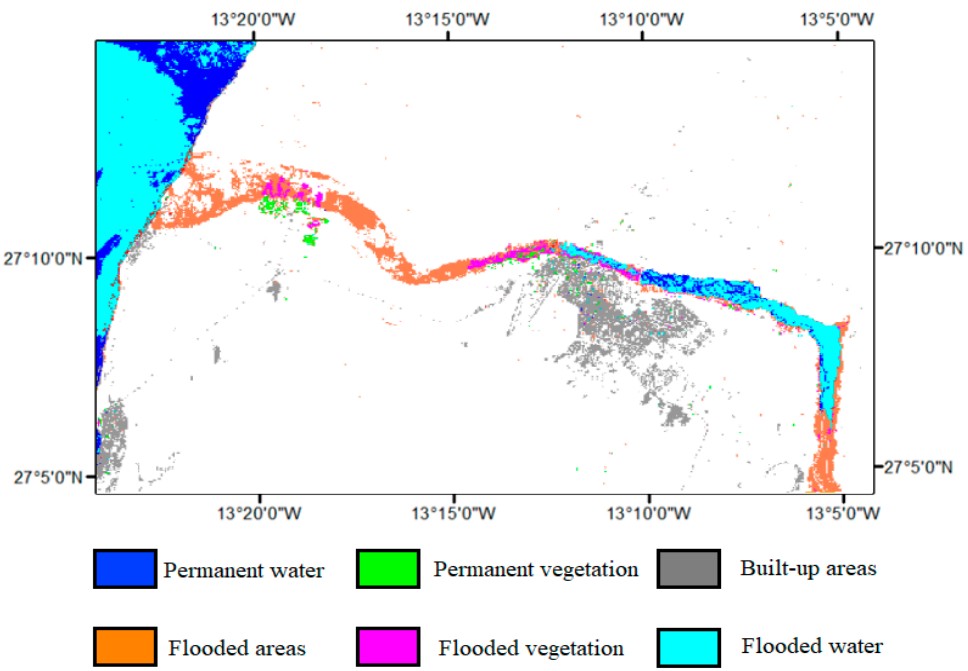

**Figure 17.** Decision tree classifier's post-classification change detection model results using both dates SVM's resultant maps (20 and 30 October 2016).

The resulting maps of the SVM classification for both dates were integrated into the constructed decision tree as inputs in order to compare the class before and after the event for each pixel, then a description for it was decided on, as explained in the tree of Figure 6.

This technique also enabled us to remove the clouds and shadows, being a temporary class, as fortunately they were not present at the acquisition time of the first scene (20 October 2016). The results of the decision tree model are shown in the Figure 17. The added details can be seen on the map, as we mentioned, when we compare the cloudy date from the SVM classification, and the clouds and their shadows has been successfully removed by the rules of the constructed decision tree. Another detail is that, on this map, we can see three types of flooded areas: (1) flooded areas mainly consisting of bare soils, sands, and built-up areas that have been overflowed by water; (2) flooded water that can be explained as the permanent water that has been invaded by flooded water containing clay, which also gives an idea of the situation before and the extent of the water in the dam; and (3) flooded vegetation indicating the great damage to agricultural lands feeding the neighboring city of Laayoune, as half of them were destroyed by the overflow. A comparative mapping of both approaches is shown in Figure 18. It shows an acceptable consistency between the flooding tasks observed through change-detection-reclassified satellite images and predictions by HEC-RAS on areas with regular slopes, whether at the city level or at the cornice.

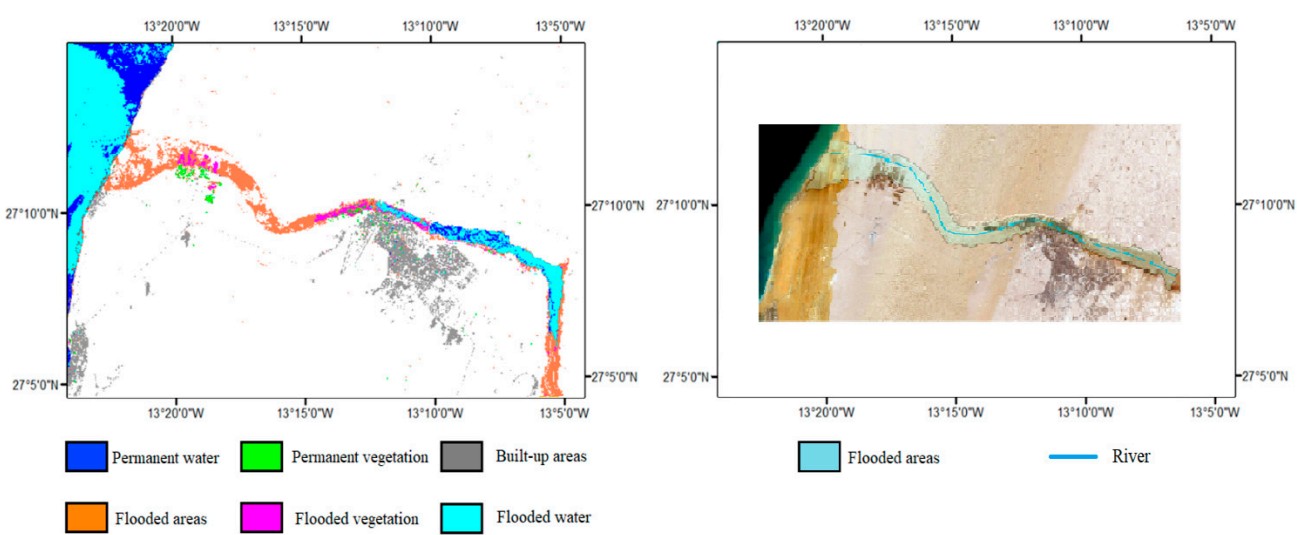

**Figure 18.** Comparison of remotely sensed and hydraulic-based simulation of the flash flood maps.

## 4. Conclusions

The purpose of this study is to evaluate the flash flood that happened in late 2016 in the southern Moroccan city of Laayoune. This work was accomplished using a two-pronged strategy: a hydraulic approach based on models implemented in WMS and HEC-RAS and a remote sensing approach based on classification and change detection techniques applied to Sentinel 2 satellite images from the European Space Agency. Through the use of the hydraulic study, the delineation of the watershed and the physical features of the flow were simulated, and the return period was forecasted to behave virtually identically in the future, with the minimal expansion of the riverbanks. The water level can rise as high as 13 m, inundating hundreds of hectares of neighboring plains in the northern portion of the city of Laayoune and agricultural regions near Foum El Oued due to the high-flow velocity. Before and after the occurrence, the SVM classifier was employed to map land cover and land use. The overall accuracy (Kappa coefficient) was 94.41 percent (0.91), and 87.33 percent (0.81), respectively for both dates, when compared to the ground-truth data. The decision tree was built with the maps produced by the SVM classification for both dates as inputs, producing a change detection map with increased detail. The remote sensing technology has enabled us to monitor the damage that has been done to the area following the catastrophe with details on the buildings affected, farms flooded, and the extent of the river. As Sentinel-2 has a 5-day revisit interval, a fundamental constraint of the technique is that the satellite overpass time does not always occur the day after the events. In the future, however, this might be improved by launching additional satellites with higher temporal resolution, or for the time being, by combining two or more satellites in the hope of capturing the after-event. The results of this study, despite being interesting and promising, could be enhanced with the inclusion of reanalysis data, such as the latest ERA5-Land product. The results could be refined with the assistance of more experimental instruments, which are scarce in Saharan regions. Future use of satellite images with a higher resolution could also enhance the scientific value of this study.

**Author Contributions:** Conceptualization, E.-A.N., B.S., E.H.B., A.M., N.-E.L. and A.L.; Data curation, E.-A.N. and B.S.; Formal analysis, E.-A.N., B.S., E.H.B., A.M., N.-E.L. and A.L.; Investigation, E.-A.N.; Methodology, E.-A.N., B.S. and E.H.B.; Software, E.-A.N., B.S. and E.H.B.; Validation, E.-A.N. and B.S.; Visualization, E.-A.N. and B.S.; Writing - original draft, E.-A.N., B.S. and A.M.; Writing - review & editing, E.-A.N., B.S. and E.H.B.; Project administration, N.-E.L. and A.L. All authors have read and agreed to the published version of the manuscript.

**Funding:** This research received no external funding.

**Data Availability Statement:** The data presented in this study are available in this article.

**Acknowledgments:** The authors would like to thank the Laboratory of Molecular and Ecophysiological Modeling of Department of Physics, faculty of science at Semlalia Cadi Ayyad University Marrakech, the watershed agency of Laayoune Saquia El Hamra Oued Eddahab (ABHSHOD), and the regional directorate of meteorology (DRM) Laayoune, Morocco.

**Conflicts of Interest:** The authors declare no conflict of interest.

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
