# Peer review of "Hydraulic Modeling and Remote Sensing Monitoring of Floodhazard in Arid Environments—A Case Study of Laayoune City in Saquia El Hamra Watershed Southern Morocco"

_water, doi:10.3390/w14213582_

Round 1
Reviewer 1 Report
Hydraulic Modelling and Remote Sensing Monitoring of 2 Floodhazard in Arid Environments-A Case Study of Laayoune city in Saquia El Hamra Watershed Southern Morocco.
Overview
The objective of this work is to monitor wetlands after a flash flood in the Saharan arid region of Saquia El Hamra, Morocco, using a technique that combines hydraulic modeling and remote sensing, employing WMS, Hec Ras, Support Vector Machines, and a decision tree. The work is interesting, it could be useful in areas vulnerable to floods, however, the work suffers from some shortcomings that should be improved prior to publication.
Recommendations:
Here are some recommendations:
1. Line 92-97 The location of the stations where the information on flows, precipitation and temperature was collected should be indicated with coordinates and their location on a map.
2. Line 213 What criteria were applied to select the sites to determine ground truth? Where are they located? What are the characteristics of each site in terms of land use and type of soil? Include a map
3. If there is a dam in the study area, the process for including it in the river model should be specified.
4. Figure 7 should include terrain elements, e.g., the existing dam, should be clearly indicated.
5. The characteristic values of the basin that are calculated and presented in lines 232 - 234 are not used in any additional calculation, although they are informative, the need to include them is not justified if they are not used in the development of the research.
6. Hec-Ras is applied to determine floodplains, but a map of the results is not presented, which should be included.
7. Having hydraulic information obtained through the model, it would be interesting to discuss elements such as flow types, flow velocities, flow rates, water mirror width, among others.
8. The effect that the dam had on the flood event should be analyzed, did it contribute to mitigate the flood, did the dam suffer damage from the flood, etc.
9. Line 275-276. What are the evaporation levels in the study area to support the statement that "The presence of clay in flooded water helped distinguishing the flooded areas”?
10. Spectrally what happens in the reflectivity of dry clay and wet clay?
11. Line 267-282 It is not discussed why the overall accuracy and kappa coefficient of the classifications performed are reduced.
12. When applying the decision tree with the previously made classifications, it could not be considered in itself as a method to define flooded areas since its efficiency will depend on the accuracy of the classifications made, Perhaps the decision tree should be rethought based on the spectral characteristics of the images before and after flooding.
13. There is no comparative analysis of the flood zones obtained by Hec-Ras and those obtained by remote sensing, this comparison should be made cartographically (extent obtained in each case), areas in km2 that have been flooded and the Kappa index could be calculated to quantify the differences between them.
14. The conclusions should include the limitations of the study and future lines of development.
Reviewer 2 Report
A brief summary
The aim of the study is to monitor wetland areas after a flash-flood event in arid region Saquia El hamra Saharan of Morocco, using a technique that combines hydraulic modelling and remote sensing technology, namely satellite images. The hydrological parameters of the watershed were determined by the WMS software. Flood flow was modelled and simulated using HEC HMS and HEC-RAS software. To map the flooded areas, two satellite images (Sentinel-2 optical images) taken before and after the event were used. Three classifications were carried out using two powerful classifiers: Support Vector Machines and Decision Tree. The first classifier was applied on both dates’ images, and the resulting maps were used as input for a constructed decision tree model as a post-classification change detection process.
Comments and Suggestions for Authors:
1. The application of HEC-HMS model is not found in the paper. I suggest authors to provide more details on the how it is coupled to HEC-RAS and satellite images.
2. What is the novelty of this study. This has to be highlighted in the last part of the introduction section.
3. Two Models settings, parameters and simulation results should be presented in more detail.
4. The results and conclusion are short. It is suggested to add a discussion section, including the comparison and analysis between the research and other research results, and the innovation of the research.
Round 2
Reviewer 1 Report
The comments made have been satisfactorily received. I recommend the publication of the work.